# Knee Joint Osteoarthritis in Overweight Cats: The Clinical and Radiographic Findings

**DOI:** 10.3390/ani13152427

**Published:** 2023-07-27

**Authors:** Joanna Bonecka, Michał Skibniewski, Paweł Zep, Małgorzata Domino

**Affiliations:** 1Department of Small Animal Diseases and Clinic, Institute of Veterinary Medicine, Warsaw University of Life Sciences (WULS-SGGW), 02-787 Warsaw, Poland; 2Department of Morphological Sciences, Institute of Veterinary Medicine, Warsaw University of Life Sciences (WULS-SGGW), 02-787 Warsaw, Poland; 3OchWET Veterinary Clinic, 02-119 Warszawa, Poland; 4Department of Large Animal Diseases and Clinic, Institute of Veterinary Medicine, Warsaw University of Life Sciences (WULS-SGGW), 02-787 Warsaw, Poland

**Keywords:** BCS, radiographic signs, severity, radiographs, feline

## Abstract

**Simple Summary:**

Osteoarthritis (OA) is a common condition among cats. It is characterized by progressive degenerative joint disease. OA results from repair and degeneration of articular cartilage, in association with alterations in subchondral bone metabolism, osteophytosis, and synovial inflammation. In cats, OA is often secondary to an underlying cause, such as trauma. Cats with knee joint OA can show typical symptoms including a general reduction in activity, reluctance to jump, deterioration in appearance, and even aggression. Subtle symptoms that owners can observe are reluctance to move and apathy. Cats with knee problems may not show the typical clinical symptoms of lameness. After excluding other causes of disturbing symptoms, further diagnostic imaging is advised to visualize typical radiographic signs of OA. Thus, visualization of osteophytes, enthesophytes, effusion, soft tissue swelling, subchondral sclerosis, and intra-articular mineralization can be used to score the severity of OA. This study aimed to investigate the occurrence of clinical symptoms and radiological signs of knee joint OA in cats and to assess their prevalence concerning cats’ body condition scores. Radiographic imaging of the knee joints of 64 cats was performed, and considered signs were compared between underweight, normal-weight, and overweight groups. Severe feline knee joint OA appears with similar frequency in underweight, normal-weight, and overweight cats. Therefore, regardless of the cat’s body weight, when the owner reports any unusual behavior of the cat, the veterinarian should take a detailed history driven to identify non-specific clinical symptoms of OA.

**Abstract:**

Despite a high prevalence of osteoarthritis (OA) reported in the domesticated cat population, studies on feline knee joint OA are scarcer. Knee joint OA is a painful, age-related, chronic degenerative joint disease that significantly affects cats’ activity and quality of life. In dogs and humans, one may consider overweight as a risk factor for the development and progression of knee joint OA; therefore, this study aims to assess the severity of knee joint OA in the body-weight-related groups of cats concerning clinical symptoms and radiographic signs. The study was conducted on sixty-four (*n* = 64) cats with confirmed OA. The demographic data on sex, neutering, age, and breed were collected. Then, the body condition score (BCS) was assessed, and each cat was allocated to the underweight, normal-weight, or overweight group. Within clinical symptoms, joint pain, joint swelling, joint deformities, lameness, reluctance to move, and apathy were graded. Based on the radiographic signs, minor OA, mild OA, moderate OA, and severe OA were scored. Prevalence and co-occurrence of the studied variables were then assessed. Joint pain was elicited in 20–31% of the OA-affected joints, joint deformities in 21–30%, and lameness in 20–54%, with no differences between weight-related groups. Severe OA was detected in 10–16% of the OA-affected joints, with no differences between weight-related groups. Severe OA in feline knee joints appears with similar frequency in overweight, underweight, and normal-weight cats. However, the general prevalence of clinical symptoms and radiographic signs is different in overweight cats.

## 1. Introduction

The Osteoarthritis Research Society International (OARSI) defined osteoarthritis (OA) as a disorder affecting movable joints that manifests first as an abnormal joint tissue metabolism and next as functional derangements often leading to illness [1]. Thus, OA is considered as a common chronic degenerative form of arthritis that affects the synovial joints of humans and various species of animals [2,3]. The prevalence of OA in cats ranges from 16% to 91% [4,5,6,7,8,9], depending on the studied population. Clarke et al. [4] reported a 16% prevalence of OA in a population of 218 cats, aged from 0.2 to 18 years. Godfrey [5] showed a 22% prevalence of OA in a population of 491 cats, aged from 3 to 19 years. Hardie et al. [6] reported a 26% prevalence of appendicula joint OA in a population of 100 cats older than 12 years. Freire et al. [7] evidenced a 46% prevalence of appendicula OA in a group of 30 cats at an average age of 12 years. Slingerland et al. [8] showed a 61% prevalence of OA in a population of 100 cats older than 6 years but only 5% in knee joints. Finally, Lascelles et al. [9] noted a 91% prevalence of OA in a similar population of 100 cats, where knee joint OA was reported in 50% of affected joint OA [9]. As the prevalence of feline OA increased with age [9], currently the cat’s age is considered to be an identified OA risk factor [8,9].

Considering feline OA etiology, primary and secondary OA are distinguished. The term primary OA is used for idiopathic OA in Scottish Fold affected by osteochondrodysplasia [10] or mucopolysaccharidosis [11]. The majority of cases of primary OA have no identified initiating factor and are referred to as age-related cartilage degeneration seen in older cats [9]. On the other hand, secondary OA may be caused by congenital joint malformation, joint deformity, joint dislocation, traumatic joint injury, and hypervitaminosis A [9,12,13]. It may develop as a consequence of hip dysplasia in susceptible breeds [14,15], patellar luxation in more susceptible Abyssinian and Devon Rex cats [16,17,18], or patellar displacement, which is more common in Domestic Shorthair cats [19]. Moreover, secondary OA may appear after joint infection, as in the case of *Mycoplasma* spp. infection in polyarthritis [13], as well as after nonspecific infection when the immune response is defective [20].

One should consider that the etiology and the mechanism underlying the pathogenesis of OA are not fully understood; therefore, the treatment is focused on reducing clinical symptoms and if possible disease-modifying therapies [21]. Unlike dogs, whose lameness is the major clinical symptom of OA [22], clinical symptoms of feline OA are subtle [23,24]. Among them, the stress response is clinically the most difficult manifestation of chronic pain to recognize. The clinical symptoms of feline OA include apathy, joint pain on palpation, joint swelling, joint deformities, impaired function of the affected joint manifested by lameness, and reluctance to move [5,6,7,13,25,26,27,28]. Thus, the symptom-reducing therapy in cats is based on the administration of non-steroidal anti-inflammatory drugs (NSAIDs). However, their use in cats with stable renal insufficiency has potential drawbacks such as a higher level of proteinuria [29]. Meloxicam is the most popular, long-term, safe NSAID prescribed by veterinary surgeons and orthopedists [30]. Another NSAID, robenacoxib, is a highly selective COX-2 inhibitor developed as a painkiller and anti-inflammatory drug for dogs and cats, which can be used without adverse effects [31,32]. One may observe that OA pain is more complex than inflammatory pain alone [33]; therefore, an appropriate painkiller targeted at neuropathic pain is occasionally necessary. The study on cats with OA showed that the administration of gabapentin, a gamma-aminobutyric acid (GABA) receptor agonist, improves the cat’s activity levels [34]. Another study on old cats with OA demonstrated desirable painkilling outcomes after the administration of tramadol, an opioid receptor agonist [33]. A recent study in 2021, introduced frunevetmab, the feline monoclonal antibody therapy, into the treatment of chronic pain in cats with OA and reported significant improvement in cats’ behavior [35,36]. However, one may postulate that cats disguise part of the clinical symptoms of disease due to the instinct for self-preservation [37], due to the solitary and territorial innate behavior of cats. Therefore, the treatment strategy for cats with OA should consider the pharmacological symptom-reducing therapy with anti-inflammatory or analgetic drugs used in combination with joint supplementation, and rehabilitation. Surgery should be considered the last resort [9,28,38,39].

OA manifests itself in cartilage degradation, increase in cartilage cell metabolism, synovial inflammation, hyperplasia, and hypertrophy as well as bone changes, of which bone changes are the most reliably identifiable radiologically [4,5,6,9,27]. Thus, radiographic imaging enables the diagnosis of OA by exposing the radiographic signs of degenerative changes in the affected joint. Therefore, the use of coherent terminology and standardized nomenclature is necessary for the proper OA recognition and the feasible assessment of disease severity and progression. One may observe that the degenerative changes in the feline knee joints are radiologically characterized by the presence of the narrow and irregular joint space (thin and uneven lucency between the adjacent cortical bones); osteophytes and enthesiophytes (bone outgrowths on the surface of cortical bone); subchondral bone cyst (the osteolytic area of well-delimited increased lucency, localized in the cortical and subcortical bone) and/or subchondral bone sclerosis (the area of increased opacity within the cortical and subcortical bone); periosteal proliferation (the area of mildly increased opacity outside the cortical bone), and intra-articular mineralization (the area of severely increased opacity inside joint space) [4,5,13,15,26,40,41,42,43]. Assessment of the particular radiographic signs allows for the determination of disease severity considering the clinical symptoms.

We hypothesized that in the feline knee joint, severe OA is more frequent in overweight than in underweight and normal-weight cats and is associated with more severe clinical symptoms. Thus, the objectives of the present study were: (1) to assess the severity of knee joint OA in the overweight, underweight, and normal-weight groups of cats concerning cats’ age and sex; (2) to explore whether overweight is associated with more severe clinical symptoms; (3) to explore whether overweight is associated with OA severity.

## 2. Materials and Methods

### 2.1. Study Design

A study was conducted on six hundred and sixty-two (*n* = 662) cats presented for radiological examination of pelvic limbs. Examinations were performed between 2019 and 2022 in the Small Animal Clinic at the Institute of Veterinary Medicine at the Warsaw University of Life Sciences. All cats represented privately owned clinical patients. Owners provided written consent for the cat’s inclusion in the study. All the performed procedures represented the standard diagnostic tests; therefore, no ethical approval was required. The type of the study was categorized as prospective, longitudinal, and observational.

### 2.2. Clinical Data Collection

All cats underwent an initial examination that included the collection of medical history as well as general physical and detailed orthopedic examinations. In the medical history, data on sex, neutering, age, and breed as well as reluctance to move and apathy were obtained from each cat. The reluctance to move and apathy were assessed by the owner in the cat’s own environment. The reluctance to move was considered present when the owner reported that the cat did not want to move even when the owner encouraged the cat to move with a treat. The apathy was considered present when the cat manifested a lack of interest or concern compared to the previous, normal behavior of this cat. Body weight was measured using an electronic scale (Veterinary Scale for small animal practice, Bielskie Wagi, Bielsko-Biala, Poland). The body condition score (BCS) was used to assign cats to groups, as the body weight standard varies between cat breeds. A nine-point scale of the BCS was used to assess underweight (1–4 points), normal weight (5 points), and overweight (6–9 points) following Teng et al. [44]. The BCS was rated by two independent researchers by palpation of subcutaneous fat, visual assessment of the bone location and shape, and the waist of each cat. Then, the result was presented as a mean value from two measurements.

An orthopedic evaluation of the pelvic limbs was performed following Lascelles et al. [45]. The orthopedic evaluation consisted of observation and careful palpation of both knee joints, with each cat being assessed by the same experienced observer (J.B.). The observation was performed with the cat freely moving around the examination room. The palpation was performed with the cat in lateral recumbency, using minimal restraint provided by a single assistant (P.Z.). The same order was followed in every cat for the evaluation (passive observation, active observation, palpation). During the orthopedic evaluation, the lack (0) or presence (1) of local joint pain on palpation, joint swelling, joint deformities, and impaired function of the affected joint manifested by lameness was assessed. The joint pain was detected manually by palpation and considered present when withdrawal from manipulation, resists, body tenses, vocalization/increase in vocalization, orientation to the site, hissing, biting, escaping/preventing manipulation, and/or marked guarding of the palpated area were noted. The joint swelling was detected manually by palpation and considered present when an abnormal enlargement of a knee joint, typical for the accumulation of fluid, was noted. The joint deformities were detected manually by palpation and considered present when the alteration of the physiological bony shape of a knee joint was noted. The lameness was detected by passive observation and considered present when the gait or stance of a cat was abnormal. The result of the observation was confronted with the medical history data provided by the owner.

Initial inclusion criteria were based on a documented history consistent with OA and the presence of at least one clinical symptom of OA. Exclusion criteria were based on the history and radiographic signs of knee joint neoplasia, acute trauma, and/or luxation.

### 2.3. Radiological Data Collection

For each cat, the mediolateral projection for both knee joints was achieved. Following Lascelles et al. [9] radiographs were centered on the midpoint of the knee joint. A focus-table distance of 90 cm was used, with an exposure of 50 kVp and 2.5–3.2 mAs for the mediolateral view of the knee joints. Radiography continued until good-quality projection of the knee joint was obtained. Quality control was performed by the experienced observer (J.B.). Radiographs were taken using an X-ray system CPI Indico IQ (Communications & Power Industries Canada Inc., Georgetown, Canada). The radiographs were acquired on the computer in DICOM format using Ubuntu software (Canonical Ltd. Ubuntu Foundation, Isle of Man, Great Britain) and assessed using an advanced DICOM viewer Ginkgo CADx (GNU Lesser General Public License). The radiographic examination and image evaluation followed the international guidelines for small animal diagnostic imaging [46,47,48].

### 2.4. Data Processing

The severity of radiographic signs of knee joint OA was assessed using a scale (0–4) proposed by Lascelles et al. [9] for feline OA scoring expended by the details proposed by Kellgren and Lawrence [49] for the human knee joint OA scoring. Within the radiographic signs, the width and shape of the joint space, cortical bone surface with the presence of osteophytes and/or enthesiophytes, subchondral bone pattern with the presence of subchondral bone cyst and/sclerosis, periosteal proliferation, and intra-articular mineralization were scored as summarized in Table 1. The exemplary radiographs of scores 0–4 are presented in Figure 1.

### 2.5. Statistical Analyses

For all the statistical analyses performed, the level of significance was set at *p* < 0.05, and GraphPad Prism6 software was used (GraphPad Software Inc., San Diego, CA, USA).

#### 2.5.1. Descriptive Statistics

For statistical analysis, the data were divided into groups regarding cats’ sex, neutering, breed, and age. Each cat was also assessed using BCS in a different group as follows the underweight (UW), normal-weight (NW), or overweight (OW) group. To describe the sex, neutering, and breed groups, a total number of cats was used. For describing the age characteristics of the studied group, the minimum and maximum values, as well as mean±SD were used. The sex structure of studied groups was compared between the UW, NW, and OW groups, and total group separately using the chi-square test. Percentage entering values were used for this test due to different group sizes. The overall severity of knee joint OA and the age structure of the studied groups were tested for normality using the Shapiro–Wilk test for each group separately. At least one data series was not normally distributed; thus, the age values and sex type occurrence were presented using the median and interquartile range, and compared between groups using the Kruskal–Wallis test. When a significant difference was evidenced, the post hoc Dunn’s multiple comparisons test was used.

#### 2.5.2. Prevalence of Symptoms and Signs of Knee Joint OA

The occurrence of each clinical symptom was annotated as 0 when not present and as 1 when present. The severity of knee joint OA was annotated as shown in Table 1. The prevalence of clinical symptoms and radiographic signs of knee joint OA was calculated for the UW, NW, OW, and total groups, separately. For comparison between the UW, NW, OW, and total groups, the chi-square test with the percentage entering values due to different group sizes was used. The occurrence of each symptom or sign was compared between groups using the Kruskal–Wallis test. When a significant difference was evidenced, the post hoc Dunn’s multiple comparisons test was used. The prevalence was presented using the number of cats and the percentage (%) of each symptom or sign in each group, separately.

## 3. Results

### 3.1. Descriptive Statistics Results

Cats with a recent history of knee joint neoplasia (*n* = 41), acute trauma (*n* = 14), and/or luxation (*n* = 14) were excluded from the study. Cats with no radiographic signs of knee joint OA (*n* = 536) were excluded from the study. Finally, sixty-four (*n* = 64) cats with confirmed OA were enrolled in the study. The studied cats’ group included 39 females and 25 males, aged between 1 and 20 years (mean±SD: 11.2±4.8). Within those subgroups, 16 females and 13 males were neutered. Cats represented different breeds, namely 30 European Domestic Shorthair cats, 7 British Shorthair cats, 5 Devon Rex cats, 5 Maine Coon cats, 4 Ragdoll cats, 3 Bengal cats, 2 Abyssinian cats, 2 Nebelung cats, 2 Scottish Fold cats, 1 American Curl cat, 1 Cornish Rex cat, 1 Persian cat, and 1 Siberian cat. All of these cats presented radiographic signs of OA.

The UW group included 8 females and 3 males, aged between 10 and 17 years (mean±SD: 14.1±2.5). Within those subgroups, 2 females and 1 male were neutered. All cats represented the European Domestic Shorthair breed. The NW group included 20 females and 15 males, aged between 1 and 20 years (mean±SD: 11.1±5.2). Within those subgroups, 5 females and 6 males were neutered. These cats represented different breeds, namely 16 European Domestic Shorthair cats, 7 British Shorthair cats, 5 Devon Rex cats, 3 Bengal cats, 2 Abyssinian cats, 1 American Curl cat, and 1 Cornish Rex cat. The OW group included 11 females and 8 males, aged between 4 and 20 years (mean±SD: 9.9±4.5). Within those subgroups, 9 females and 8 males were neutered. These cats represented different breeds, namely 5 Maine Coon cats, 4 Ragdoll cats, 4 European Domestic Shorthair cats, 2 Nebelung cats, 2 Scottish Fold cats, 1 Persian cat, and 1 Siberian cat.

The overall severity of knee joint osteoarthritis (OA) did not differ between the UW, NW, and OW groups (*p* = 0.19) (Figure 2A). Similarly, the age of examined cats did not differ between the UW, NW, and OW groups (*p* = 0.07) (Figure 2B). The sex structure of examined cats differed between groups as shown in Table 2. The sex structure of the OW group differed with the UW (*p* < 0.0001), NW (*p* < 0.0001), and total groups (*p* < 0.0001), respectively. However, no differences were found between the UW and NW groups (*p* = 0.53), UW and total groups (*p* = 0.45), as well as the NW and total groups (*p* = 0.38). In the OW group, the occurrence of females was lower (*p* = 0.01) (Figure 2C), and the occurrence of neutered females was higher (*p* = 0.03) (Figure 2D) than in the UW and NW groups. No differences in the occurrence of males (*p* = 0.30)(Figure 2E) and neutered males (*p* = 0.30) (Figure 2F) were noted between the studied groups.

### 3.2. Prevalence of Symptoms and Signs of Knee Joint OA

The prevalence of the clinical symptoms of knee joint OA differed between groups as shown in Table 3. The prevalence of the clinical symptoms in the UW group differed from the NW (*p* = 0.0004), total (*p* = 0.0002), and OW (*p* = 0.0005) groups, respectively. The prevalence of the clinical symptoms in the NW group differed with total (*p* < 0.0001) and OW (*p* < 0.0001) groups, respectively. The prevalence of the clinical symptoms in the total group also differed from the OW group (*p* < 0.0001).

Concerning the consecutive clinical symptoms of knee joint OA, one may observe that joint swelling was not observed in the UW and OW groups. Among other clinical symptoms of knee joint OA, only the occurrence of apathy was higher in the NW group than in the UW and OW groups (*p* = 0.003) (Figure 3F). No differences between studied groups were found for joint pain (*p* = 0.63) (Figure 3A), joint deformities (*p* = 0.86) (Figure 3B), joint deformities (*p* = 0.05) (Figure 3C), reluctance to move (*p* = 0.20) (Figure 3D), and sum of clinical symptoms (*p* = 0.41) (Figure 3F).

The prevalence of the radiographic signs of knee joint OA differed between groups as shown in Table 4. The prevalence of the radiographic signs in the UW group differed with the OW group (*p* = 0.01) but not with the NW (*p* = 0.06) and total (*p* = 0.60). No other differences in the prevalence of the radiographic signs were found. However, concerning the consecutive radiographic signs of knee joint OA, one may observe that no differences were found in the occurrence of minor OA (*p* = 0.06 (Figure 4A), mild OA (*p* = 0.42) (Figure 4B), moderate OA (*p* = 0.19) (Figure 4C), and severe OA (*p* = 0.87) (Figure 4D) between the studied groups.

## 4. Discussion

To highlight the most relevant results of the current study, one may observe that the overall severity of knee joint OA did not differ between body-weight-related groups of cats. It is worth noting that the previous studies [9,45] have not compared the prevalence of the clinical symptoms and radiographic signs of feline knee joint OA depending on body weight; thus, such results are presented in the current study for the first time. Although the prevalence of the clinical symptoms and radiographic signs was different in the OW group than in other groups, the specific symptom-related difference remains unidentified. Thus, the hypothesis of the more frequent occurrence of severe knee joint OA in overweight cats can be rejected. The current study fills the gap in the research on feline knee joint OA. Lascelles et al. [9] found a very high prevalence of appendicular and axial skeleton OA in cats, and ever since OA has been considered the most common orthopedic disease of domesticated cats. Previous studies suggested the elbow [5,6,50,51] and hip [4,9,18,25] joints are the feline joints most commonly affected by OA. However, Lascelles et al. [9] reported that in the domesticated cat population, the most frequently affected joints are the hip, followed by the knee, tarsus, and then elbow [9]. Despite this important observation, studies on feline knee joint OA are scarce.

Lascelles et al. [9] evaluated the prevalence of radiographic signs of OA for the association with patient demographics, including age, body weight, sex, BCS, % time spent indoors/outdoors, vaccination status, and diet. Lascelles et al. [9] considered age to be the most essential variable that affected OA prevalence in cats, so, after accounting for age, body weight and BCS were found to not be significantly related to OA. In the Lascelles et al. [9] study, mean body weight was 5.1±1.6 (range, 2.1–10.3 kg), and median BCS was 3 (range, 1–5). In the later study, Lascelles et al. [45] evaluated the relationship between radiographic signs of feline OA, clinical symptoms, and joint goniometry. Within demographic data, age, weight, BCS, and sex were considered. In the Lascelles et al. [45] study, mean body weight was 5.1±1.6 (range, 2.1–10.2 kg), and median BCS was 3 (range, 1–5); thus, in [9,45], the same group of 100 cats was studied. Lascelles et al. [45] confirmed age as the most essential variable that affected the relationship between the occurrence of OA and clinical symptoms, so body weight and BCS did not change the significance of the relationship between the clinical symptoms and OA. In the current study, the overall OA severity was similar in underweight, normal-weight, and overweight cats; thus, those results are consistent with the previous studies on cats [9,45]. On the other hand, in dogs, a link between overweight and OA has been evidenced [52]. Similarly, in humans, overweight is considered to be a risk factor for the development and progression of knee joint OA; thus, weight loss is advised as the first line of the OA treatment strategy both in dogs [53] and humans [54,55,56]. In a case of steady weight accumulation, the opportunity to improve OA outcomes is missed [57]. One may conclude that in cats, the OA treatment strategy should consider pharmacological symptom-reducing therapy, joint supplementation, and rehabilitation [9,28,38,39], regardless of the cat’s body weight.

One may state that in the studied group, overweight occurs more frequently in neutered females. This result is consistent with the previous study showing the association between cats’ overweight, breed, sex, age, and neutering status [58]. However, in previous studies, contradictory to the current one, the male sex was associated with an increased risk of being overweight [58,59]. Instead, all the studies agree that neutered cats are predisposed to being overweight due to an increase in daily food intake, decrease in metabolic rate, and decrease in activity [58,59,60]. However, the increased risk of OA in neutered cats has not been evidenced yet, unlike neutered dogs where the OA-neutering status relationship was confirmed, but the underlying mechanism is still not fully understood [61]. Öhlund et al. [58] reported a decreased risk of being overweight in Birman and Persian breeds; however, the authors did not identify any particular cat breed at an increased risk of being overweight. In the current study, in the OW group, the Maine coon and Ragdoll cats prevailed, and all Nebelung, Scottish Fold, Persian, and Siberian cats were included. The NW group contained mostly a prevalence European Domestic Shorthair cats as well as all British Shorthair, Devon Rex, Bengal, Abyssinian, American Curl, and Cornish Rex cats.

On the other hand, in the UW group, only European Domestic Shorthair cats were included. This last observation is contradictory to the previous studies, as when comparing purebred and European Domestic Shorthair cats, the European cats were more often overweight [58,62]. To summarize breed-related factors, one may say that cat breeds can not be advised as a risk factor for being overweight [59] since no genetic factors predisposing particular cat breeds to overweight have been proven [63], whereas dog breeds, such as Golden Retriever, Pug, Beagle, English Springer Spaniel, and Border Terrier, have been proven to be predisposed to overweight [64]. It is worth noting that among considered demographic factors, age is so far the only identified risk factor affecting both prevalences of OA [8,9] and worsening of clinical symptoms of OA [45]. However, old-aged cats are less predisposed to being overweight compared with middle-aged cats [58,59,62]. The highest likelihood of being overweight was exhibited for the middle-aged cats (5 to 11 years old) [59,65] and middle-aged dogs (3 to 11 years old) [64], which are at lower risk of OA [9,52]. However, in the current study, no differences were found in the age of the cats between the body-weight-related groups. Thus, the age-related impact on the severity of knee joint OA is difficult to assess and discuss.

Scarlett and Donoghue [66] showed that overweight cats were 2.9 times as likely to be taken to veterinarians because of lameness than normal-weight cats. However, the authors just presumed that the evidenced lameness was related to OA and soft-tissue injuries. Later studies did not confirm the relationship between clinical symptoms and radiographic signs of OA and overweight [45], while the relationship between clinical symptoms and radiographic signs of OA was described in detail. Therefore, the lack of differences in lameness between the UW, NW, and OW groups evidenced in the current study may be considered consistent with the recent Lascelles et al. [45] study. Lascelles et al. [45] showed the elbow and hip joints affected by OA are most frequently found to be painful, followed by the knee and tarsus joints. The authors found joint pain in between 21% and 22% of OA-affected knee joints. Similarly, in the current study, joint pain appeared in 20%, 31%, and 21% of OA-affected knee joints in the UW, NW, and OW groups, respectively. Moreover, no differences were found between these groups. Lascelles et al. [45] also observed that the elbow joint affected by OA is most frequently found to show crepitus, effusion, and thickening followed by the knee and tarsus joints. The authors evidenced crepitus in between 10% and 14% of OA-affected knee joints, effusion in between 12% and 13% of OA-affected knee joints, and thickening in between 14% and 17% of OA-affected knee joints. As crepitus was not investigated in the current study, the appearance of joint deformities but not joint swelling ranged from 21% to 30%, with no differences between studied groups. One may note that in the current study, no joint swelling was observed in the UW or OW groups. Lascelles et al. [45] showed that the range of motion is significantly decreased in knee joints affected by OA. However, the authors reported the values of the studied range rather than the percentage of joints with a lower range of motion. In the current study, the motion decrease in OA-affected knee joints was not assessed. Lascelles et al. [45] suggested that the absence of clinical symptoms, such as pain, crepitus, effusion, and thickening of the affected joint, could be used to rule out OA. However, in the recent study, the authors did not consider non-specific symptoms, such as apathy, which seems to be the most common clinical symptom. Therefore, one may conclude that when the owner reports any unusual behavior of the cat, the veterinarian should take a detailed history driven to identify non-specific clinical symptoms of OA, especially as dogs suffering from knee joint OA most frequently show discomfort followed by other more specific clinical symptoms such as pain, limited joint range of motion, loss of muscle mass, reduced activity level, and lameness [53,67,68]. Similarly, in humans suffering from knee joint OA, joint pain and tenderness, short-term morning stiffness, and restricted movement may occur early in the disease. Crepitations, bone enlargement, and decreased range of motion suggest mild to moderate OA, whereas severe OA is characterized by pain, muscle wasting, and deformities [69].

As the future directions in the prevention of feline OA, the detailed evaluation of any behavioral changes, clinical symptoms, and cats’ housing conditions should be considered. One can not rule out stress as a factor initiating the onset of OA changes in joints in cats [70,71,72]. Thus, further research should include the assessment of stress indicators or the collection of data on stress behaviors in history. The imaging diagnostics of feline knee joint OA should be extended by using more accurate modalities such as computed tomography [51,73] and magnetic resonance imaging [74], as it is in humans [75,76] and dogs [77]. Although conventional radiography is still the first-choice imaging modality in the diagnosis and follow-up of feline OA [9,78], its significant drawbacks such as low sensitivity and poor correlation with clinical status support the dissemination of CT- and MRI-based OA diagnosis [79].

## 5. Conclusions

Severe feline knee joint OA appears with similar frequency in overweight, underweight, and normal-weight cats. However, the prevalence of clinical symptoms and radiographic signs is different in overweight cats, which affects neutered females more often. Concerning consecutive symptoms and signs, it can not be unequivocally stated that the course of the disease in overweight cats is more severe. Therefore, regardless of the cat’s body weight, when the owner reports any unusual behavior of the cat or changes in its behavior, the veterinarian should take a detailed history to identify non-specific clinical symptoms of OA. Thus, the veterinarian may decide to perform a radiographic examination to identify radiographic signs of OA, knowing that non-specific symptomatic cats require constant radiological monitoring to diagnose OA as early as possible and to initiate symptomatic treatment to improve their quality of life.

## Figures and Tables

**Figure 1 animals-13-02427-f001:**
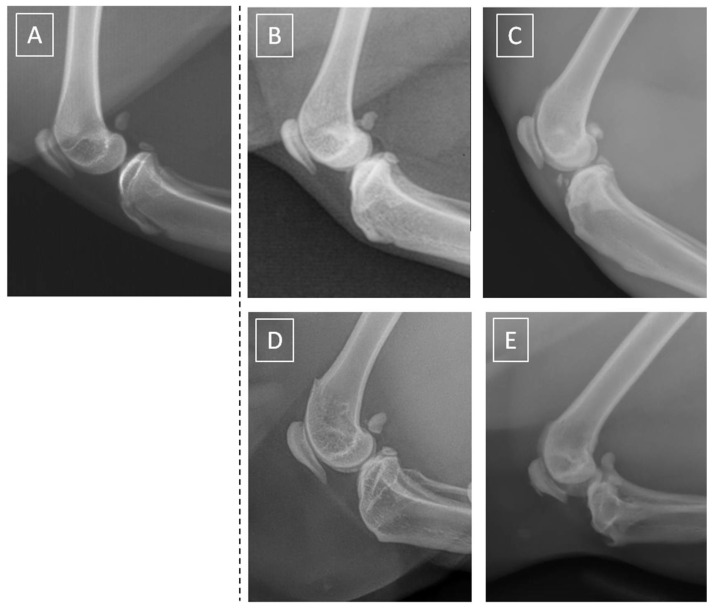
Exemplary mediolateral view radiographs of feline knee joints scored as normal (**A**), minor OA (**B**), mild OA (**C**), moderate OA (**D**), and severe OA (**E**). The dashed line separates the image of a normal cat knee from the images of OA knees.

**Figure 2 animals-13-02427-f002:**
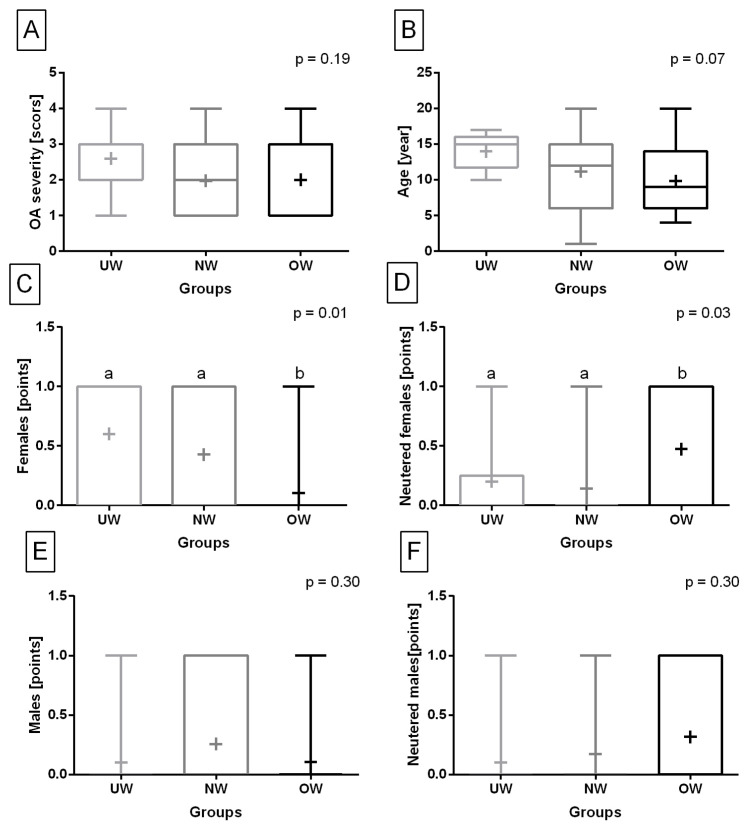
The overall severity of knee joint osteoarthritis (OA) (**A**), age (**B**), as well as the occurrence of females (**C**), neutered females (**D**), males (**E**), and neutered males (**F**) in the underweight (UW), normal-weight (NW), and overweight (OW) groups. Data in box plots are represented by the lower quartile, median, and upper quartile, whereas whiskers represent minimum and maximum values. Additionally, the mean values are marked by “+”. Lowercase letters indicate differences between groups for *p* < 0.05.

**Figure 3 animals-13-02427-f003:**
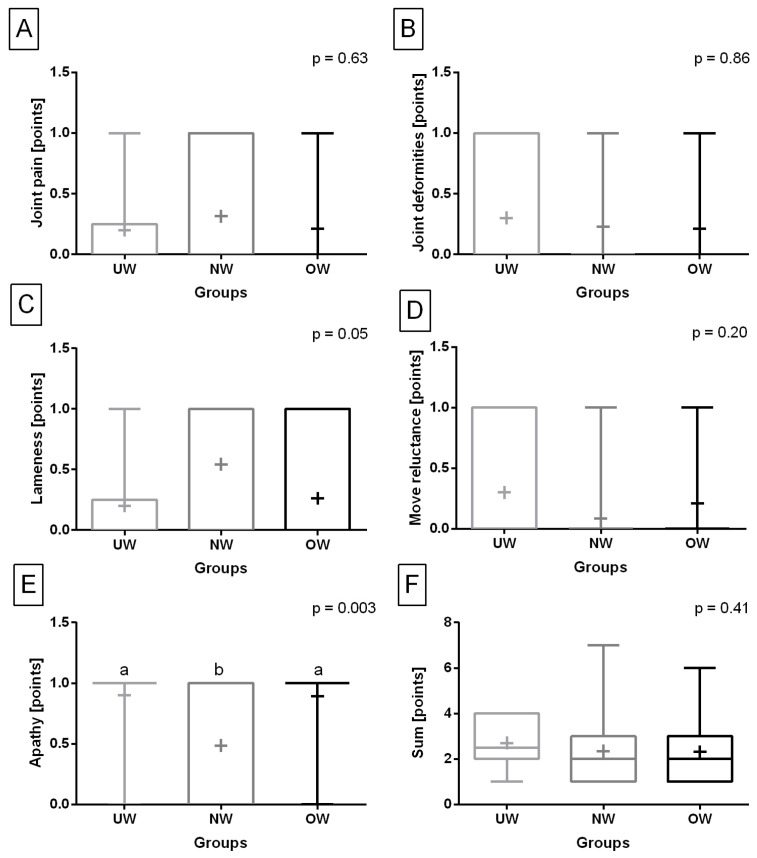
Occurrence of joint pain (**A**), joint deformities (**B**), lameness (**C**), reluctance to move (**D**), and apathy (**E**), as well as the sum of clinical symptoms (**F**) of knee joint osteoarthritis (OA) in the underweight (UW), normal-weight (NW), and overweight (OW) groups. Data in box plots are represented by the lower quartile, median, and upper quartile, whereas whiskers represent minimum and maximum values. Additionally, the mean values are marked by “+”. Lowercase letters indicate differences between groups for *p* < 0.05.

**Figure 4 animals-13-02427-f004:**
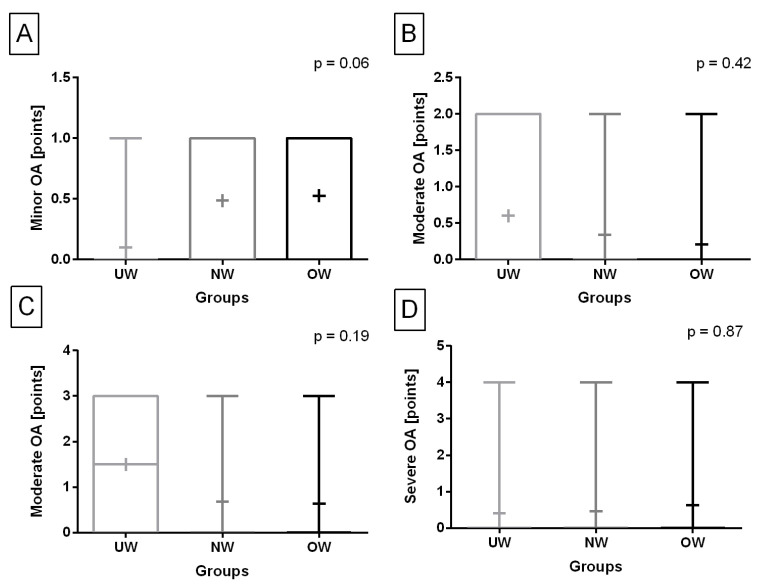
Occurrence of minor (**A**), mild (**B**), moderate (**C**), and severe (**D**) radiographic signs of knee joint osteoarthritis (OA) in the underweight, normal-weight, and overweight groups. Data in box plots are represented by the lower quartile, median, and upper quartile, whereas whiskers represent minimum and maximum values. Additionally, the mean values are marked by “+”.

**Table 1 animals-13-02427-t001:** Radiographic signs used for scoring the severity of knee joint osteoarthritis (OA).

Score	Severity	Radiographic Signs
0	normal	normal width and shape of the joint space; smooth cortical bone surface; normal subchondral bone pattern; no periosteal proliferation; no intra-articular mineralization
1	minor	normal width and normal or irregular joint space; normal or irregular cortical bone surface; smooth subchondral bone pattern; flat periosteal proliferation; minor intra-articular mineralization
2	mild	narrow and irregular joint space with osteophytes; irregular cortical bone surface with well-defined protuberance; smooth subchondral bone pattern; flat periosteal proliferation; mild intra-articular mineralization
3	moderate	narrow and irregular joint space with multiple osteophytes, enthesiophytes, and marked asymmetry; irregular cortical bone surface with well-defined bone proliferation; subchondral bone cyst; flat periosteal proliferation; moderate intra-articular mineralization
4	severe	completely narrow joint space with large osteophytes and enthesiophytes; severe deformation of cortical bone surface; subchondral bone sclerosis; flat or intense periosteal proliferation; severe intra-articular mineralization

**Table 2 animals-13-02427-t002:** The sex structure of underweight (UW), normal-weight (NW), overweight (OW), and total groups. Data are represented as the number and (%) of cats representing each considered sex/neutered sex. Differences between groups are considered significant for *p* < 0.05.

Variables	UW	NW	OW	Total
Number of cats	10	35	19	64
Females	6 (60%)	15 (43%)	2 (11%)	23 (36%)
Neutered females	2 (20%)	5 (14%)	9 (47%)	16 (25%)
Males	1 (10%)	9 (26%)	2 (11%)	12 (19%)
Neutered males	1 (10%)	6 (17%)	6 (32%)	13 (20%)
Chi-square test				
UW		*p* = 0.53	*p* < 0.0001	*p* = 0.45
NW			*p* < 0.0001	*p* = 0.38
OW				*p* < 0.0001

**Table 3 animals-13-02427-t003:** Prevalence of the clinical symptoms of knee joint osteoarthritis (OA) in underweight (UW), normal-weight (NW), overweight (OW), and total groups. Data are represented as the number and (%) of cats representing each symptom. For the sum of clinical symptoms, the medians and (interquartile ranges) are presented. Differences between groups are considered significant for *p* < 0.05.

Variables	UW	NW	OW	Total
Number of cats	10	35	19	64
Clinical symptoms				
Joint pain	2 (20%)	11 (31%)	4 (21%)	17 (27%)
Joint swelling	0 (0%)	3 (9%)	0 (0%)	3 (5%)
Joint deformities	3 (30%)	8 (23%)	4 (21%)	15 (23%)
Lameness	2 (20%)	19 (54%)	5 (26%)	26 (41%)
Reluctance to move	3 (30%)	3 (9%)	4 (21%)	10 (16%)
Apathy	9 (90%)	17 (49%)	17 (89%)	43 (67%)
Sum	2.5 (2; 4)	2 (1; 3)	2 (1; 3)	2 (1; 3)
Chi-square test				
UW		*p* = 0.0004	*p* = 0.0002	*p* = 0.0005
NW			*p* < 0.0001	*p* < 0.0001
OW				*p* < 0.0001

**Table 4 animals-13-02427-t004:** Prevalence of the radiographic signs of knee joint osteoarthritis (OA) all in underweight (UW), normal-weight (NW), overweight (OW), and total groups. Data are represented as the number and (%) of cats representing each sign. Differences between groups are considered significant for *p* < 0.05.

Variables	UW	NW	OW	Total
Number of cats	10	35	19	64
Radiographic signs				
Minor OA (score 1)	1 (10%)	17 (49%)	10 (53%)	28 (44%)
Mild OA (score 2)	3 (30%)	6 (17%)	2 (11%)	11 (17%)
Moderate OA (score 3)	5 (50%)	8 (23%)	4 (21%)	17 (27%)
Severe OA (score 4)	1 (10%)	4 (11%)	3 (16%)	8 (13%)
Chi-square test				
UW		*p* = 0.06	*p* = 0.01	*p* = 0.13
NW			*p* = 0.60	*p* = 0.46
OW				*p* = 0.76

## Data Availability

The data presented in this study are available on request from the corresponding author.

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
