# Peer review of "Knee Joint Osteoarthritis in Overweight Cats: The Clinical and Radiographic Findings"

_animals, 2023, doi:10.3390/ani13152427_

Round 1

Reviewer 1 Report

The aims of this study were:

to assess the severity of knee joint OA in the overweight, underweight, and normal weight groups of cats concerning cats’ age and sex;

to explore whether the overweight is associated with more severe clinical symptoms;

to explore  whether overweight is associated with OA severity.

I suggest adding a section in material and methods to explain the clinical signs detected. It is mandatory to improve the scientifical value of the paper to discuss the relationship between the data intensely.

It is hard for the reader to understand the concept line of the experiment.

The authors refer to joint deformity, to local temperature increase but do not explain the lesion detected.

What about the exclusion criteria?

What about previous or concomitant orthopaedic conditions?

The Authors refer to reluctance to move but do not give any detail about lumbosacral junction conditions.

I guess it is essential to address all these topics to improve the manuscript substantially.

Moreover, the discussion of results seems confusing and too long.

L 131-134: please rephrase the sentences; this section should be reported in the results

L 136: change in mediolateral

L 145: radiographic symptoms are not correct. I suggest changing in radiographic signs

L 146: Why did the Authors refer to a human scale if, in the veterinary literature, different scales are proposed to score the stifle joint OA?

L 169: refer to the scale to shorten the section

L 172: please, rephrase the sentence referring to the clinical symptoms and the radiographical signs (in all the text)

L 173: all cat

L 180: The sum of clinical symptoms refers to 0-1, presence or absence?

But the clinical signs investigated are different in severity and clinical relevance.

This way of managing the data is not appropriate and does not give the right relevance. A 4-point scale to assess the severity of clinical presentation should probably be necessary.

L 186: Please, change in female and male (in all the text).

Please, also provide English editing and a profound revision of the manuscript.

Author Response

Dear Reviewer,

We would like to thank you for all your insights and the time invested to look at our manuscript. We are grateful for pointing out several limitations in our study design and insufficiency of the English quality. We made every effort to address all your comments and complete the extensive and significant revisions of the manuscript. We hope that as a result of the changes, the manuscript has gained sufficient scientific value. Thank you very much for your time and detailed review. Best regards.     

Comment R.1.1

I suggest adding a section in material and methods to explain the clinical signs detected. It is mandatory to improve the scientifical value of the paper to discuss the relationship between the data intensely. It is hard for the reader to understand the concept line of the experiment. The authors refer to joint deformity, to local temperature increase but do not explain the lesion detected. What about the exclusion criteria? What about previous or concomitant orthopaedic conditions? The Authors refer to reluctance to move but do not give any detail about lumbosacral junction conditions. I guess it is essential to address all these topics to improve the manuscript substantially.

Answer R.1.1

Thank you very much for this recommendation. Following your suggestion the new subsection has been separated. Every effort has been made to improve the concept line of the manuscript by correcting incorrect syntax. The lesion detection has been explained in detail. The orthopedic examination protocol was adapted from Lascelles et al. (2012). The exclusion criteria were listed at the end of this subsection.

Comment R.1.2

Moreover, the discussion of results seems confusing and too long.

Answer R.1.2

Thank you for this remark. Please, see that the discussion section was designed following the "common standard format" that usually considers the following points:

  1. One sentence summary that highlights the most relevant results.
  2. A thorough discussion of each result obtained concerning the corresponding study objective: was the tested hypothesis confirmed or not? What previous evidence supports the specific result or not? It is critical to compare/contrast the result obtained with previous literature in the feline species first, then in veterinary medicine, and finally in human medicine (if not enough data are available for comparison in veterinary medicine
  3. Future directions
  4. Conclusions

We hope this "common standard format" is completed.

Comment R.1.3

L 131-134: please rephrase the sentences; this section should be reported in the results

Answer R.1.3

Thank you for this recommendation. The paragraph has been moved.

Comment R.1.4

L 136: change in mediolateral

Answer R.1.4

Thank you for pointing it out. The change has been made throughout the whole manuscript.

Comment R.1.5

L 145: radiographic symptoms are not correct. I suggest changing in radiographic signs

Answer R.1.5

Thank you for pointing it out. The change has been made throughout the whole manuscript.

Comment R.1.6

L 146: Why did the Authors refer to a human scale if, in the veterinary literature, different scales are proposed to score the stifle joint OA?

Answer R.1.6

Many thanks for this question. The knee joint OA is underestimated in cats and at present the primary causes are not completely known. The only two researchers who described feline knee joint OA are Lascelles et al. (2010) and Lascelles et al. (2012). Please, see the scale (0–4) proposed by Lascelles et al. (2010) is poorly described. Therefore, we decide to add details described and widely known in humans. Following your remark, the description in the respective paragraph has been expanded.

Comment R.1.7

L 169: refer to the scale to shorten the section

Answer R.1.7

Thank you for this recommendation. The section has been shortened.

Comment R.1.8

L 172: please, rephrase the sentence referring to the clinical symptoms and the radiographical signs (in all the text)

Answer R.1.8

Thank you for this recommendation. It has been corrected throughout the whole manuscript.

Comment R.1.9

L 173: all cat

Answer R.1.9

Many thanks for pointing it out. Following the Reviewer's 2 comments the All cat group was renamed.

Comment R.1.10

L 180: The sum of clinical symptoms refers to 0-1, presence or absence? But the clinical signs investigated are different in severity and clinical relevance. This way of managing the data is not appropriate and does not give the right relevance. A 4-point scale to assess the severity of clinical presentation should probably be necessary.

Answer R.1.10

Thank you very much for this important remark. We agree a 4-point scale of the clinical symptoms will be more accurate. However, we cannot retrospectively evaluate and re-evaluate raw data. Certainly, taking into account your significant doubt, we will expand our next studies to assess the severity of clinical symptoms. Since this is no longer possible in this work, the results should be presented as preliminary.

Comment R.1.11

L 186: Please, change in female and male (in all the text).

Answer R.1.11

Many thanks for this recommendation. It has been done.

Comment R.1.12

Please, also provide English editing and a profound revision of the manuscript.

Answer R.1.12

Thank you for this recommendation. The manuscript has been revised by a native English speaker colleague. We hope the language has been improved. Please, see after acceptance, each paper in the MDPI publisher is additionally checked and corrected by an expert in scientific writing, and this is the standard protocol of the Animals Journal publication. As we are not native English speaker experts in scientific writing, we like and value this quality very much at MDPI publisher.

Reviewer 2 Report

The study aims to investigate the predisposing factors and clinical signs of knee osteoarthritis in cats. The OA is underestimated in cats and at present the primary causes are not completely known. The work is quite well structured, but in my opinion, changes can be made to make it clearer.

Here are some comments:

Line 12: “Radiological symptoms” is a mistake, we are not talking about symptoms but about signs. I suggest correcting it because it is repeated many times in the text.

Line 54: correct Scottish fold

Line 121: The BCS was used to assign cats to groups, as the body weight standard varies between cat breeds

Line 136: I am not sure that only the mid-lateral projection is sufficient to assess osteoarthritis in a joint district. Why this choice? how were the projections performed? In the images, the positioning does not seem standardized.

Line 156: suggest using acronyms for the different groups.

Line 160: I recommend changing the name of the group 'all cats' can generate misunderstandings and is not suitable for a scientific article

Lines 155-186: these sections should be rewritten and made more comprehensible. Revision of the language is essential.

Line 235: In my opinion, the main result of the study should be emphasized more. Add and discuss the bibliography

Lines 250-256: a sentence is repeated twice

Concepts are often unclear or difficult to understand due to incorrect syntax. I suggest a revision of the English language by an expert in scientific writing. 

Author Response

Dear Reviewer,

We would like to thank you for all your insights, time, and detailed review of our manuscript. We are very pleased with the opinion the work is quite well structured. We made every effort to address all remarked changes to make the article clearer. After all those corrections, we see a significant improvement in the quality of the manuscript. Thus thank you very much and best regards.

Comment R.2.1

Line 12: “Radiological symptoms” is a mistake, we are not talking about symptoms but about signs. I suggest correcting it because it is repeated many times in the text.

Answer R.2.1

Thank you for pointing it out. The mistake has been corrected throughout the whole manuscript.

Comment R.2.2

Line 54: correct Scottish fold

Answer R.2.2

Thank you for pointing it out. It has been corrected.

Comment R.2.3

Line 121: The BCS was used to assign cats to groups, as the body weight standard varies between cat breeds

Answer R.2.3

Many thanks for this correction. It has been addressed.

Comment R.2.4

Line 136: I am not sure that only the mid-lateral projection is sufficient to assess osteoarthritis in a joint district. Why this choice? how were the projections performed? In the images, the positioning does not seem standardized.

Answer R.2.4

Thank you very much for this question. In the case of 90.6% of cats, two orthogonal projections were performed following Lascelles et al. (2010, 2012). However, in 6 cats no anterior-posterior image was taken. Based on the lateral image, those cats were assigned to the minor group OA (4 cats) and mild OA group (2 cats). We decided to leave the evaluation criteria on the side picture so as not to exclude these six cats, because it is very difficult and time-consuming to gather a database of OA cats large enough to present the results of the research. We did not want to reduce the size of the entire study group (64 cats). In addition, in all other cases, except for the OA staging (assignment to a given group), the lateral and AP images were the same, so the choice of projection did not affect the classification of individual cats. This observation confirmed our belief not to reduce the size of the group, but to limit the description of the methodology.

We have also improved the description of the data collection. Following Lascelles et al. (2010) radiographs were centered on the midpoint of the knee joint. The described positioning is standard in the Small Animals Clinic of the Institute of Veterinary Medicine, Warsaw University of Life Sciences. Radiography continued until good-quality projection of the knee joint was obtained. Quality control was performed by the lead investigator (J.B.). Please, see Lascelles et al. (2010, 2012) did not report any imaging settings, whereas, in the case of our study, the best quality images of the knee joint was achieved by the reported settings.

Comment R.2.5

Line 156: suggest using acronyms for the different groups.

Answer R.2.5

Many thanks for this suggestion. The acronyms have been introduced.

Comment R.2.6

Line 160: I recommend changing the name of the group 'all cats' can generate misunderstandings and is not suitable for a scientific article

Answer R.2.6

Many thanks for this recommendation. The group was named "total" and corrected throughout the whole manuscript.

Comment R.2.7

Lines 155-186: these sections should be rewritten and made more comprehensible. Revision of the language is essential.

Answer R.2.7

Thank you very much for this remark. The section has been rewritten.

Comment R.2.8

Line 235: In my opinion, the main result of the study should be emphasized more. Add and discuss the bibliography

Answer R.2.8

Thank you very much for this recommendation. It is worth noting that the previous studies of Lascelles et al. (2010, 2012) have not compared the prevalence of the signs of feline knee joint OA depending on body weight, thus such results are presented in the current study for the first time. Following your recommendation it has been highlighted and supported by the appropriate references.

Please, see that the discussion section was designed in accordance with the "common standard format" that usually considers the following points:

  1. One sentence summary that highlights the most relevant results.
  2. A thorough discussion of each result obtained in relation to the corresponding study objective: was the tested hypothesis confirmed or not? What previous evidence supports the specific result or not? It is critical to compare/contrast the result obtained with previous literature in the feline species first, then in veterinary medicine, and finally in human medicine (if not enough data are available for comparison in veterinary medicine
  3. Future directions
  4. Conclusions

We hope this "common standard format" is completed.

Comment R.2.9

Lines 250-256: a sentence is repeated twice

Answer R.2.9

Thank you very much for this remark. We checked carefully LL 250-256 and did not find repeated sentences. Please, see Lascelles et al. published two very similar papers (2010 and 2012) which are cited here one by one. The description of Lascelles et al. (2010) and Lascelles et al. (2012)  research is similar, however, addresses two studies.

Comment R.2.10

Concepts are often unclear or difficult to understand due to incorrect syntax. I suggest a revision of the English language by an expert in scientific writing.

Answer R.2.10

Thank you for this recommendation. Every effort has been made to improve the concept line of the manuscript by correcting incorrect syntax. The manuscript has been revised by a native English speaker colleague. We hope the language has been improved. Please, see after acceptance, each paper in the MDPI publisher is additionally checked and corrected by an expert in scientific writing, and this is the standard protocol of the Animals Journal publication. As we are not native English speakers experts in scientific writing, we like and value this quality very much at MDPI publisher.

Reviewer 3 Report

I consider that the article is well written and structured. The methodology is clear and the results support the conclusions.

Perhaps table 2, 3 and 4 do not understand the statistical significance part very well. 

It would be good if the radiographic images and graphs were of a larger size.

In the body of the article, BCS is mentioned for the first time on line 121, and not on 123. Therefore, the abbreviation should be explained on line 121 and on line 123 only the abbreviation should be included.

Line 124: I suggest saying “…following Teng et al., [44]” instead of “…following [44]”

Author Response

Dear Reviewer,

We would like to thank you for all your comments and time invested in looking at our manuscript. We are pleased with your opinion that the article is well-written and structured, the methodology is clear, and the results support the conclusions. We hope that we have addressed all raised issues. Thank you very much and best regards. 

Comment R.3.1

Perhaps table 2, 3 and 4 do not understand the statistical significance part very well.

Answer R.3.1

Thank you for expressing this insecurity. Please see that in Tables 2-4 each group is compared to other groups. To avoid doubling the same result, p values for half of the table are presented. For example, the p-value is shown for NW in a column and UW in a row but not for NW in a row and UW in a column because they are the same values. The p-value for UW in a row and UW in a column is also not shown because the same groups were not compared with each other.

Comment R.3.2

It would be good if the radiographic images and graphs were of a larger size.

Answer R.3.1

Thank you for this recommendation. The images on each figure have been enlarged.

Comment R.3.3

In the body of the article, BCS is mentioned for the first time on line 121, and not on 123. Therefore, the abbreviation should be explained on line 121 and on line 123 only the abbreviation should be included.

Answer R.3.3

Thank you for pointing it out, the abbreviation has been corrected.

Comment R.3.4

Line 124: I suggest saying “…following Teng et al., [44]” instead of “…following [44]”

Answer R.3.4

Many thanks for this remark, it has been addressed.

Round 2

Reviewer 1 Report

Second round revision

L 132: observer, not assessor

L 136-138: To be honest, the detection of temperature increase for the diagnosis of OA is too difficult to accept. It can be reliable in human beings and in acute joint injuries but not for animals with chronic conditions and hair. I suggest removing the sentences.

L 143-144: when considering bone deformities, I suggest speaking about the alteration of physiological bony shape and not to enlargement. Please, rephrase the sentences.

L145-146: In my opinion, it is not reliable to detect the reluctance to move in some cats outside their own environment. I suggest taking this information in anamnesis. Please, rephrase the sentences and refer them to the owner's report.

L148-149: Did you use an articular goniometry? Please stated if a goniometer was used. If it was not, rephrase the sentences because the single visualization is not useful to assess the ROM reduction. To better say you cannot assess the variation degree of ROM.

L 149: Please, be more precise about apathy and reluctance to move. Seems to be the same condition.

L 154-155: did the joint deformities reason for the exclusion?

I do not understand if the deformity was investigated as a possible end stage of OA. Please, focus on the topics.

Joint deformities due to OA are a condition possible in human beings and not in cats at the OA end stage.

I suggest a deep language revision and editing

Author Response

Dear Reviewer,

We would like to thank you for all your insights and the time invested to look at our manuscript. We agree the evaluation of the clinical symptoms could be confused in the case of knee joint OA, thus we made every effort to rebuild the results and discussion section following all your suggestions. Please, see the response to your comments/questions attached below. We hope after those improvements our manuscript will be of sufficient quality to be published in the Animals. Thank you very much and best regards.

Comment R.1.1

L 132: observer, not assessor

Answer R.1.1

Many thanks for pointing it out. It has been corrected.

Comment R.1.2

L 136-138: To be honest, the detection of temperature increase for the diagnosis of OA is too difficult to accept. It can be reliable in human beings and in acute joint injuries but not for animals with chronic conditions and hair. I suggest removing the sentences.

Answer R.1.2

We can not fully agree that the assessment of the increased local temperature can not be conducted in the case of animals since it is a routine part of the orthopedical examination. However, to avoid confusion, all results concerning the increase in local temperature have been removed.

Comment R.1.3

L 143-144: when considering bone deformities, I suggest speaking about the alteration of physiological bony shape and not to enlargement. Please, rephrase the sentences.

Answer R.1.3

Thank you very much for this recommendation. The sentence has been rephrased.

Comment R.1.4

L145-146: In my opinion, it is not reliable to detect the reluctance to move in some cats outside their own environment. I suggest taking this information in anamnesis. Please, rephrase the sentences and refer them to the owner's report.

Answer R.1.4

Thank you very much for this recommendation. Following this and the next remarks, in the material and method section, the reluctance to move and apathy were moved to the owner's report and clarified.

Comment R.1.5

L148-149: Did you use an articular goniometry? Please stated if a goniometer was used. If it was not, rephrase the sentences because the single visualization is not useful to assess the ROM reduction. To better say you cannot assess the variation degree of ROM.

Answer R.1.5

Thank you very much for this remark. In this research, the goniometer was used. Therefore to avoid confusion, all results concerning the decreased range of motion have been removed.

Comment R.1.6

L 149: Please, be more precise about apathy and reluctance to move. Seems to be the same condition.

Answer R.1.6

Thank you very much for this recommendation. Following this and the previous remarks, in the material and method section, the reluctance to move and apathy were moved to the owner's report and clarified.

Comment R.1.7

L 154-155: did the joint deformities reason for the exclusion?

Answer R.1.7

No, the deformities were not a reason for the exclusion. The reason for the exclusion was the history and radiographic signs of knee joint neoplasia, acute trauma, and/or luxation.

Comment R.1.8

I do not understand if the deformity was investigated as a possible end stage of OA. Please, focus on the topics. Joint deformities due to OA are a condition possible in human beings and not in cats at the OA end stage.

Answer R.1.8

Many thanks for this comment. As you highlighted in the first round of reviews, The aims of this study were: to assess the severity of knee joint OA in the overweight, underweight, and normal weight groups of cats concerning cats’ age and sex; to explore whether the overweight is associated with more severe clinical symptoms; and to explore whether overweight is associated with OA severity. Thus there was no comparison in the prevalence of clinical symptoms severity between the initial (minor OA), early intermediate (minor OA), late intermediate (moderate OA), and end (severe OA) stages of the disease. We agree with your opinion, that we should just focus on the topic, thus this additional comparison has not been provided. In the revised version of the manuscript, the deformity is not investigated as a possible end stage of OA.

Comment R.1.9

I suggest a deep language revision and editing.

Answer R.1.9

Thank you very much for this recommendation. We asked another English native speaker to read the article and introduce the language. All are marked in the text. According to the language editor, the language of the article is not perfect, but it is quite decent and understandable. We have made every effort to improve the substantive content and language of the article and we very much hope that we have managed to address all your remarks. Once again, thank you very much, and best regards.

Reviewer 2 Report

The authors made the changes according to suggestions. I think the article is suitable for publication. 

Reviewer 3 Report

Suggestions have been included and the manuscript is of sufficient quality to be published.

Round 3

Reviewer 1 Report

The Manuscript after the revision process is improved significantly. I consider it suitable for publication.